Association between bone marrow adipose tissue, abdominal adipose tissue distribution, and volumetric bone mineral density in the Chinese adult population: a retrospective cohort study

Zhu Wei 1
Guan Wen-Min 1
Chen Bo-Xin 2
Lu Yi-Da 2
Li Jia 1
Yuan Xiao-Qing 1
Li Wei-Hua 1
Yu Feng-Xia 1
Liu Jing-Yi 2
Yin Hong-Xia 1 3
Wei Lin 2
Wang Zhen-Chang 1
Zhu Zhi-Jun zhu-zhijun@outlook.com 2
Zhang Peng zpqz1021@foxmail.com 1
1 Department of Radiology, Beijing Friendship Hospital, Capital Medical University , Beijing , China
2 Liver Transplantation Center, National Clinical Research Center for Digestive Diseases, Beijing Friendship Hospital, Capital Medical University , Beijing , China
3 Department of Medical Engineering, Beijing Friendship Hospital, Capital Medical University , Beijing , China
Anson Lesley
Electronic publication date: 2025 Dec 4
Publication date: 2025
Volume: 13
Electronic Location ID: e20446
Received 2025 May 16; Accepted 2025 Oct 31
Copyright: ©2025 Zhu et al.
Copyright year: 2025
Copyright holder: Zhu et al.
License: This is an open access article distributed under the terms of the Creative Commons Attribution License, which permits unrestricted use, distribution, reproduction and adaptation in any medium and for any purpose provided that it is properly attributed. For attribution, the original author(s), title, publication source (PeerJ) and either DOI or URL of the article must be cited.
License URL: https://creativecommons.org/licenses/by/4.0/

Keywords: Quantitative computed tomography, Volumetric bone mineral density, Osteoporosis, Bone marrow adipose tissue, The proton density fat fraction

Funding: National Natural Science Foundation of China 82302182 Beijing Science and Technology Plan Project Z241100009024020 Tongzhou District Science and Technology Innovation Talent Support Project CXTD2024007 This work was supported by National Natural Science Foundation of China (82302182), Beijing Science and Technology Plan Project (grant No. Z241100009024020) and Tongzhou District Science and Technology Innovation Talent Support Project (CXTD2024007). The funders had no role in study design, data collection and analysis, decision to publish, or preparation of the manuscript.

==============================
Background

Increasing studies have suggested that bone closely interacts with bone marrow adipose tissue (BMAT) and abdominal adipose tissue. However, this relationship remains debated. This study aimed to explore the association among BMAT, abdominal adipose tissue, and lumbar volumetric bone mineral density (vBMD).

Methods

A total of 306 Chinese adult living liver transplant donors were enrolled in this cross-sectional study. vBMD and abdominal adipose tissue in terms of total adipose tissue (TAT), visceral adipose tissue (VAT) and subcutaneous adipose tissue (SAT), were derived using quantitative computed tomography (QCT). Chemical shift encoded magnetic resonance imaging (MRI) of the lumbar spine was performed, and proton density fat fraction (PDFF) maps were calculated. Correlation analysis and multiple linear regression were used to assess the associations among BMAT, abdominal adipose tissue distribution, and vBMD.

Results

The mean age of the participants was 38.1 ± 9.5 years (range: 19–66 years). The vBMD was significantly negatively associated with age (r = −0.508, p < 0.001), VAT (r = −0.235, p < 0.001), TAT (r = −0.127, p = 0.03), and BMAT measured by PDFF (r = −0.642, p < 0.001). After adjusting for age, gender, and BMI, multiple linear regression analysis showed BMAT (β = −0.523, p < 0.001), SAT (β = 0.116, p = 0.045) and VAT (β = −0.108, p = 0.037) to be independent factors influencing vBMD.

Conclusion

Our results indicated a negative association between BMAT and BMD. The association between abdominal adipose tissue distribution and BMD was dual; there was a positive association between SAT and BMD and a negative association between VAT and BMD. These findings suggest that the distribution of abdominal adipose tissue, particularly visceral adipose tissue accumulation, should be prioritized over total body weight in considerations of bone health.

Background

Osteoporosis leads to a decrease in bone mass and weakening of the bone microarchitecture, resulting in increased fragility and a higher risk of fracture (Anonymous, 1993). Fractures can severely weaken the quality of life of older adults and increase healthcare costs. Therefore, osteoporosis has emerged as a critical public health concern among older populations (NIH Consensus Development Panel on Osteoporosis Prevention, Diagnosis, and Therapy, 2001).

Previous studies have shown a complex and controversial relationship between abdominal adipose tissue, bone marrow adipose tissue (BMAT), and bone health (Gkastaris et al., 2020; Shen et al., 2012b). Osteoblasts and adipocytes share a common origin in mesenchymal stem cells (MSCs), and a rise in adipogenesis might compromise osteogenesis (Lanske & Rosen, 2017; Beresford et al., 1992). In parallel to the argument, several recent studies have reported that osteoporosis is associated with an increased prevalence of adiposity within the bone marrow. Additionally, numerous studies have suggested that the distribution of adipose tissue may have a more detrimental effect on bone health than increased general adiposity (Ibrahim, 2010). Aging and menopause have been linked to a shift in adipose tissue redistribution, moving fat from the extremities to the trunk and causing abnormal or excessive abdominal fat accumulation (Greendale et al., 2019). Nevertheless, the association between abdominal adipose tissue distribution and bone mineral density (BMD) remains inconsistent, and there is limited information on how the distribution of adipose tissue, notably in the visceral (VAT) and subcutaneous (SAT) compartments, affects bone fragility (Yamaguchi et al., 2009; Wang et al., 2013). A deeper insight into the relationship among adipose tissue, BMD, and BMAT is important for formulating effective osteoporosis management and treatment strategies.

Examination of the adipose tissue and BMD using diverse imaging modalities can provide significant insights into the pathophysiological processes of the musculoskeletal system. Quantitative computed tomography (QCT) to measure volumetric bone mineral density (vBMD) plays an important role in the evaluation of osteoporosis and is currently regarded as the clinical standard for vBMD measurements (Zhang et al., 2019; Ma et al., 2015). As a three-dimensional measurement method, QCT enhances the sensitivity and accuracy of BMD measurements and facilitates the acquisition of corresponding body composition information, which is widely regarded as a standard for assessing the distribution of adipose tissue and muscle (Li et al., 2013; Gao et al., 2022). Magnetic resonance imaging (MRI) techniques, such as proton magnetic resonance spectroscopy (MRS) and chemical shift-encoded MRI (CSE-MRI), are sensitive and accurate noninvasive methods for assessing BMAT content (Li et al., 2011; Baum et al., 2015). CSE-MRI accurately evaluates fat concentration by calculating the proton density fat fraction (PDFF) in tissues.

Although osteoporosis is more common in older women, both men and women experience bone loss at a younger age that continues throughout their lives. Nevertheless, to the best of our knowledge, most studies have focuses predominantly on female demographics and have been performed with older adults. The association between abdominal adipose tissue, BMAT, and BMD in younger individuals of both sexes remains unclear. Furthermore, these interrelationships may vary across populations of different ethnicities, lifestyles, and nutritional habits. Particularly, the Chinese population has been understudied in this context. Consequently, this study aimed to investigate the association among abdominal adipose tissue, BMAT, and lumbar vBMD in an adult Chinese adult population of both sexes.

Materials & Methods

Study participants

This retrospective study was approved by the Institutional Review Board and Ethics Committee of Beijing Friendship Hospital (2024-P2-211-01). Participants’ information was de-identified before analysis. This study recruited adult living liver transplant donors who underwent transplantation surgery at the liver transplantation center of Beijing Friendship Hospital, Capital Medical University, between October 2019 and March 2024. Written informed consent was obtained from each donor before surgery. Inclusion criteria were (a) living liver transplant donors who had undergone both abdominal CT and MRI scans within one month before transplantation surgery; (b) age ≥ 18 years; (c) accessibility of demographic and clinical data. The exclusion criteria were (a) abdominal CT or MRI with insufficient image quality or did not include the inferior margin of the third lumbar vertebra; and (b) history of lumbar spine fracture, arthritis, cancer, pregnancy, or use of medications that might influence the values of vBMD (e.g., the use of cortisone or prednisone). Finally, as presented in Fig. 1, 306 living liver transplant donors who met the inclusion criteria were included in this study.

Figure 1 Flowchart of subjects selection.

QCT measurement of BMD and abdominal adipose tissue distribution

All patients underwent abdominal CT scan using the same CT scanner (Emotion16, Siemens Medical Solutions). The CT scans ranged from the diaphragmatic dome to the bilateral kidneys. Parameters for the scan were set at 120 kV, 130 mA, with a slice thickness of 1.0 mm, a pitch of 1.0 mm, a 50 cm field of view, and standard reconstruction. The QCT software package (QCT PRO 4.2; Mindways, Austin, TX, USA) was used to measure BMD and abdominal adipose tissue. A Mindways Model 4 CT phantom was used for calibration. This study solely involved the post-processing of current CT datasets, avoiding any additional radiation.

In the Chinese guidelines for the diagnostic criteria of osteoporosis with QCT, at least two lumbar vertebrae of the 1st (L1), 2nd (L2), and 3rd (L3) lumbar vertebrae were recommended for BMD measurement (Shen et al., 2004). Given the scan range of the abdominal CT scans of the participants, we chose L1-L2 for vBMD measurement. The average values of L1 and L2 were used to determine the mean vBMD. The midplane of the L1-L2 vertebral body was chosen to place elliptical regions of interests (ROIs) to avoid the effects of the cortical bone and proliferative osteophytes. The area of the ROI was approximately 250 mm2.

The mid-slice axial level of the Ll-L2 intervertebral disc was used to quantify the abdominal adipose tissue compositions (Cheng, 2018). Figure 2 illustrates that the mid-slice axial level of the Ll-L2 intervertebral disc was employed to measure adipose tissue composition, which was synchronized with BMD measurement. The SAT refers to the extra-abdominal adipose tissue area, whereas the VAT area refers to the intra-abdominal adipose tissue within the abdominal cavity, bordered by the rectus, external oblique, lumbar quadrate, and psoas muscles. Semi-automatic measurements of the SAT and VAT areas were conducted using the “Tissue Composition Module” of the QCT PRO software. Total adipose tissue (TAT) area was determined by adding SAT and VAT. All ROI measurements were conducted by two trained radiologist with more than 2 years of experiences who were blinded to the participants’ general information. A random subset of 60 participants was selected to assess inter-observer reliability, which was evaluated using intraclass correlation coefficients (ICC). The ICCs for measurements were excellent (ICC > 0.90), demonstrating high reproducibility. Any discrepancies in measurements were adjudicated by a senior radiologist with 11 years of experience in this field.

Figure 2 ROI delineation for the measurement of vBMD and abdominal adipose tissue distribution.

BMAT measurement by MRI

All participants underwent a chemical shift-encoded sequence called the Iterative Decomposition of water and fat with Echo asymmetry and Least Square Estimation-iron quantification (IDEAL-IQ) using a SIGNA Pioneer 3.0 T MRI system (GE Healthcare, Milwaukee, WI, USA). A 16-channel phased-array body coil was used for acquisition. The imaging protocol included the following: repetition time (TR) 5.6 ms, echo times from 0.9 to 4.7 ms, slice thickness five mm, bandwidth 111.11 Hz, field of view 44 cm × 44 cm, matrix 140 × 160, flip angle 3° (Ma et al., 2021). After acquisition, the image data were processed on the GE workstation (Advantage Windows 4.7, GE Healthcare, Anaheim, CA, USA) to calculate the PDFF maps, which were subsequently co-registered to axial T2-weighted images. The middle plane of the vertebral body was confirmed by observing the entry of the basivertebral vein into axial sections. To evaluate the BMAT, an elliptical ROI was positioned in each vertebra (L1 to L2), with the ROI placed three times per vertebra, and the BMAT values were averaged to determine the mean PDFF of the cancellous bone (Fig. 3). All ROI measurements were conducted by two trained radiologist with more than 2 years of experiences who were blinded to the participants’ general information. Any discrepancies in measurements were adjudicated by a senior radiologist with 11 years of experience in this field.

Figure 3 Illustration of ROI placement on vertebral cancellous bone for proton density fat fraction (PDFF) measurement.

Other assessments

Standard anthropometric methods were used to measure body weight and height, and the body mass index (BMI) was calculated as weight (kg) divided by height (m) squared. Overweight was defined as having a BMI between 24 kg/m2 and less than 28 kg/m2, whereas obesity was defined as a BMI of 28 kg/m2 or higher (Zhou & Cooperative Meta-Analysis Group of the Working Group on Obesity in China, 2002). Clinical and demographic data, including age, sex, smoking habit, and alcohol intake, were collected using the hospital information system. Venous blood samples were collected after an 8-hour fast and were centrifuged within 30 min. Laboratory tests included aspartate transaminase (AST), alanine aminotransferase (ALT), alkaline phosphatase (ALP), and fasting blood glucose (GLU).

Statistical analysis

All statistical analyses were performed using SPSS software (version 22.0; SPSS, Armonk, NY, USA) and R (version 4.4.1; R Foundation for Statistical Computing, Vienna, Austria). The Kolmogorov–Smirnov test was used to test for normality. Data are summarized as mean ± SD for continuous variables and as frequency for categorical variables. Student’s t test, Fisher’s exact test, and Pearson’s chi-square test of independence were performed to assess the statistical significance between the two groups. Correlations between vBMD and age, abdominal adipose tissue, and BMAT were analyzed using Pearson’s correlation coefficients. Additionally, the relationships among different abdominal adipose tissues, BMAT, and vBMD, including age, sex, and BMI, were further tested using multiple linear regression analysis. For linear regression, the dependent variable was the average vBMD of L1-L2, and the independent variables were BMI, age, BMAT, and abdominal adipose tissue. A collinearity diagnosis was made before the multiple regression analysis. p values less than 0.05 were considered statistical significance.

Results

Demographic and clinical characteristics

The demographic characteristics of all participants are summarized in Table 1. A total of 306 participants, 156 males (51.0%) and 150 females (49.0%), were included in the analysis. The mean age was 38.1 ± 9.5 years (range: 19-66 years). The mean ages of participants in the male and female groups were 37.6 ± 9.9 years and 38.6 ± 9.1 years, respectively. The average BMI was 23.6 ± 2.9 kg/m2for all participants, with the male group at 24.1 ± 2 kg/m2 and the female group at 23.1 ± 2.8 kg/m2. The majority of male participants were classified as overweight (47.4%), whereas the majority of female participants were classified as normal weight (60.7%). Compared to the female group, the male group was more likely to be a drinker (p < 0.001) and a smoker (p < 0.001) and to have higher values of ALT (p < 0.001) and ALP (p = 0.004). There were no differences in age, GLU, or AST levels between the male and female groups (all p > 0.05).

Table 1 Characteristics of study participants (n = 306).

Variables	Total (n = 306)	Male (n = 156)	Female (n = 150)	p-value	
Age (yr)	38.1 ± 9.5	37.6 ± 9.9	38.6 ± 9.1	0.354	
BMI (kg/m2)		24.1 ± 2.9	23.1 ± 2.8	0.002**	
<18.5, n (%)	9 (3.0)	5 (3.2)	4 (2.7)		
18.5–24, n (%)	156 (51.0)	65 (41.7)	91 (60.7)		
24–28, n (%)	120 (39.0)	74 (47.4)	46 (30.7)		
>28, n (%)	21 (7.0)	12 (7.7)	9 (6)		
Smoking, n (%)	41 (13.4)	40 (25.6)	1 (0.7)	<0.001**	
Drinking, n (%)	22 (7.2)	22 (14.1)	0 (0)	<0.001**	
GLU (mmol/L)	5.2 ± 0.7	5.3 ± 0.7	5.1 ± 0.7	0.05	
AST	22.4 ± 16.0	23.3 ± 14.9	21.4 ± 16.9	0.313	
ALT	20.3 ± 14.3	24.0 ± 12.0	16.5 ± 15.5	<0.001**	
ALP	80.3 ± 25.3	84.4 ± 21.1	76.1 ± 28.5	0.004**	
SAT (cm2)	111.6 ± 57.9	88.5 ± 49.3	135.7 ± 56.6	<0.001**	
VAT (cm2)	120.0 ± 67.5	147.4 ± 72.6	91.5 ± 47.1	<0.001**	
TAT (cm2)	231.7 ± 102.5	235.9 ± 111.9	227.2 ± 92.0	0.457	
PDFF (%)	42.1 ± 10.4	44.7 ± 9.4	39.3 ± 10.7	<0.001**	
vBMD (mg/cm3)	153.1 ± 33.0	147.1 ± 30.4	159.3 ± 34.4	0.001**	
Notes.

Double asterisks (**) indicate p < 0.01.

Abbreviations BMI body mass index

GLU blood glucose

AST aspartate transferase

ALT alanine aminotransferase

ALP alkaline phosphatase

SAT subcutaneous adipose tissue

VAT visceral adipose tissue

TAT total adipose tissue

PDFF proton density fat fraction

vBMD volumetric bone mineral density

Comparison of abdominal adipose tissue, BMAT, and vBMD between males and females

Table 1 and Fig. 4 present the average values of L1-L2 vBMD, L1-L2 BMAT, and abdominal adipose tissue in males and females. The vBMD for the male group and female group were 147.1 ± 30.4 mg/cm3 and 159.3 ± 34.4 mg/cm3, repectively. BMAT was 44.7 ± 9.4% for the male group and 39.3 ± 10.7% for the female group. Compared to females, males had lower L1-L2 vBMD (p = 0.001), and higher PDFF of BMAT at L1-L2 (p < 0.001). The male group had higher VAT (p < 0.001) and lower SAT (p < 0.001). However, no significant differences in TAT were observed between two sexes.

Figure 4 The comparison of vBMD, BMAT, and abdominal adipose tissue between males and females.

As shown in Figs. 5 and 6, ageing led to vBMD decrease and PDFF increase in different age ranges between male and female groups. Prior to the age of 50, females exhibited higher vBMD and less BMAT compared to males. However, after the age of 50, females experience a dramatic decline in vBMD, resulting in lower vBMD values than their male counterparts of the same age. Similarly, women also had higher BMAT compared to men of the same age after the age of 50. The females had higher SAT and lower VAT than males of the same age in different age categories.

Figure 5 The comparison of vBMD, BMAT in different age ranges between males and females.

Figure 6 The comparison of SAT, VAT in different age ranges between males and females.

Correlations between abdominal adipose tissue distribution, PDFF of BMAT, and vBMD

As shown in Fig. 7 and Table 2, Pearson’s correlation coefficients between age, BMI, abdominal adipose tissue, and PDFF of BMAT are presented. Age showed a moderately negative correlation with vBMD (r =  − 0.508, p < 0.001). Pearson’s correlation analysis revealed that vBMD was mildly negatively correlated with VAT (r =  − 0.235, p < 0.001), and TAT (r =  − 0.127, p = 0.03). vBMD was strongly negatively correlated with BMAT (r =  − 0.642, p < 0.001). However, there was no significant correlation between the SAT and vBMD (r = 0.049, p = 0.389).

Figure 7 Scatter plot of correlation between age, BMI, PDFF, SAT, VAT, and TAT area.

Independent factors of vBMD according to the multiple linear regression model

As shown in Table 3, the relationships among different abdominal adipose tissue distributions, BMAT, and vBMD were further tested using multiple linear regression analysis with adjustment for age, sex, and BMI. The R2 and adjusted R2 values of the overall linear model were 0.545 and 0.532, respectively (p < 0.05). The model revealed that age (β = −0.326, p < 0.001), BMI (β = −0.137, p = 0.036), SAT (β = 0.116, p = 0.045), VAT (β = −0.108, p = 0.037), and BMAT (β = −0.523, p < 0.001) were significant and independent factors of vBMD.

Discussion

China has the largest and fastest-aging population worldwide. The prevalence and hazard of osteoporosis have increased significantly in recent years. Compared to the Western population, the Chinese population has distinct differences in dietary habits, nutrition, physical activity, and lifestyle. Furthermore, emerging evidence suggests that Asian populations, including Chinese individuals, may have distinct patterns of adipose tissue distribution and bone metabolism compared to Western populations (WHO Expert Consultation, 2004). Thus, in the present study, we investigated a cohort of 306 Chinese adults to better understand the Chinese population-specific association between bone and adipose tissues. The present study innovatively employed quantitative techniques, specifically QCT and IDEAL-IQ sequences, to explore the relationship among BMAT, abdominal adipose tissue distribution, and vBMD in a cohort of 306 Chinese adults. Our findings revealed that age, BMAT, VAT, and TAT were significantly and negatively correlated with vBMD. Furthermore, this study identified VAT, SAT, and BMAT as independent factors of vBMD. These findings suggest a close and interconnected relationship among abdominal adipose tissue, BMAT, and vBMD.

Relationship between BMAT and BMD

During the aging process, bone interacts with BMAT and total body fat (Duque, 2008; Rosen & Bouxsein, 2006). Previous research has shown that BMAT progressively increases with age (Dieckmeyer et al., 2015). As people age, the gradual loss of bone in the vertebral space is not simply substituted by fatty bone marrow; instead, it involves mechanisms, such as a shift in MSC differentiation, favoring adipogenesis over osteogenesis (Karampinos et al., 2015). In the current study, we observed a negative correlation between BMAT and vBMD in a Chinese adult population, which persisted even after controlling for age, sex, and BMI. This finding is in accordance with those of previous studies indicating that BMD is negatively associated with BMAT (Shen et al., 2012a). The differentiation of MSCs to either osteoblasts or adipocytes is a competing process, and the procedure is mediated by systematic and local factors. Numerous hypotheses have discussed the relationship between BMAT and bone loss. Increased BMAT reflects a passive compensation for bone loss, and fat tissue helps fill the space in the trabecular bone and may cause bone weakening. Some studies have suggested a link between BMAT and fracture (Patsch et al., 2013). Another possible mechanism is that BMAT produces adipokines and fatty acids, which directly affect bone cells. Those factors secreted by marrow fat may suppress osteoblast proliferation and promote osteoclast differentiation by inducing a lipotoxic environment for bone cells (Demontiero, Vidal & Duque, 2012; Hozumi et al., 2009). It is also known that thiazolidinediones, which activate the peroxisome proliferator-activated receptor γ, promotes adipogenesis and handicap bone formation (Lazarenko et al., 2007). Overall, the study’s outcome further substantiates the idea that the relationship between BMAT and bone is competitive.

Table 2 Correlation analysis between the abdominal adipose tissue, bone marrow adipose tissue (BMAT), age, and BMI versus vBMD.

	r values	p values	
Age (years)	−0.508	<0.001**	
BMI	−0.135	0.019*	
SAT	0.049	0.389	
VAT	−0.235	<0.001**	
TAT	−0.127	0.03*	
PDFF	−0.642	<0.001**	
Notes.

The single asterisk (*) and double asterisks (**) indicate p < 0.05 and p < 0.01, respectively.

BMI, body mass index; SAT, subcutaneous adipose tissue; VAT, visceral adipose tissue; TAT, total adipose tissue; PDFF, proton density fat fraction; vBMD, volumetric bone mineral density.

Table 3 Independent predictors of vBMD analyzed by multiple linear regression model.

	β coefficient	SE	p value	
Gender	0.08	4.071	0.899	
Age	−0.326	0.152	<0.001**	
BMI	−0.137	0.627	0.036**	
SAT	0.116	0.036	0.045*	
VAT	−0.108	0.029	0.037*	
PDFF	−0.523	0.152	<0.001**	
Notes.

The single asterisk (*) and double asterisks (**) indicate p < 0.05 and p < 0.01, respectively.

SE, standard error; BMI, body mass index; SAT, subcutaneous adipose tissue; VAT, visceral adipose tissue; PDFF, proton density fat fraction; vBMD, volumetric bone mineral density.

Relationship between different abdominal adipose tissue and BMD

Traditionally, obesity has been considered to exert a protective effect on bone quality, a notion supported by the theory that increased body mass and BMI create a mechanical load associated with increased BMD (Kim et al., 2009). Nevertheless, findings from this study, along with others investigating the associations with depot-specific body adiposity, suggest that the relationship between adiposity and BMD is more intricate. The distribution of adipose tissues appears to be significant for bone health. Studies on the relationship between VAT and SAT on bone health have yielded mixed results. The VAT and SAT are the two main categories of abdominal adipose tissue accumulation. In the present study, we observed a negative association between VAT and vBMD, and a positive association between SAT and vBMD, which is consistent with previous studies (Gilsanz et al., 2009). VAT has been reported to have strong metabolic activity and a detrimental influence on human health, which may also negatively impact bone quality (Shapses, Pop & Wang, 2017). Several mechanisms may explain the negative association between VAT and BMD. First, VAT secretes inflammatory mediators and adipokines, such as tumor necrosis factor-alpha (TNF-α) and interleukin-6 (IL-6), which can impact bone metabolism by promoting osteoclast differentiation and activation and reducing osteoclast apoptosis (Dolinková et al., 2008; Pou et al., 2007). Second, chronic inflammation in fat tissues also plays a crucial role in the development of insulin resistance (IR), which negatively impacts the proliferation and survival of osteoblasts (Pramojanee et al., 2013). Evidence has also demonstrated that greater IR is associated with lower femoral neck strength in American adults (Srikanthan et al., 2014). Furthermore, higher VAT is linked to decreased serum 25(OH)D and elevated parathyroid hormone levels, negatively affecting BMD (Bolland et al., 2006b; Bolland et al., 2006a). Zhu et al. (2020) showed that VAT mass was negatively associated with serum 25(OH)D levels in 2,223 Australian females.

Nevertheless, the association between SAT and BMD remains limited and debated, with some studies reporting a positive association and others reporting a negative or no association (Zhang et al., 2021). The divergence in these findings may be caused by different study populations or techniques used for assessing BMD and abdominal adipose tissue. In our study, the vBMD was positively associated with the SAT in both males and females. This result further supports the idea that subcutaneous fat distribution has a protective effect on bone structure and strength (Tchkonia et al., 2013). This could be explained by the leptin being predominantly produced by subcutaneous fat tissue, which may increase bone mass by stimulating osteoblast activity (Wiest et al., 2010). This finding further supports the theory that different abdominal adipose tissue distributions may have distinct effects on BMD.

This study has several strengths. First, unlike the dual-energy X-ray absorption (DXA) used in most previous studies, we took advantage of QCT in quantifing BMD and abdominal adipose tissue. While modern DXA systems can also assess BMD and body composition (Maskarinec et al., 2022), including visceral and subcutaneous adipose tissue, the QCT-based approach described in this study allows for assessment of volumetric BMD, offering greater sensitivity and accuracy than the areal BMD measurements obtained using DXA (Chen et al., 2023). The use of existing abdominal CT scans to assess BMD via QCT can provide valuable ancillary data without requiring additional radiation exposure from a dedicated scan. However, it is important to note that the effective dose of a QCT scan is higher, and DXA remains the preferred first-line clinical method for routine assessment due to its lower cost and minimal radiation. Therefore, this approach is advantageous for patients in whom CT scans are acquired for other diagnostic purposes. Second, PDFF has been proposed as a useful noninvasive tool for the accurate quantification of body fat compositions, demonstrating very good concordance with MRS or histology (Smith et al., 2014; Reeder, Hu & Sirlin, 2012). Our findings indicate that BMAT measured using CSE-MRI is negatively associated with BMD and may offer complementary information and insights into bone health assessment. Third, it is noteworthy that the most of the study population was middle-aged, with a mean age of 38.1 ± 9.5 years. Although osteoporosis is more commonly diagnosed in older women, it is crucial to recognize that bone loss is a universal issue that begins earlier in life for both sexes. Understanding the distinct mechanisms and risk factors associated with bone loss in men and women can facilitate the development of effective preventive and treatment strategies tailored to individual needs. Lastly, our results indicate that abdominal adipose tissue may exert a dual effect on BMD, whereby VAT is negatively associated with BMD, and SAT is positively associated with BMD. Unlike other non-modifiable risk factors for osteoporosis, such as age and gender, adipose tissue distribution parameters are easily accessible and can be modified through lifestyle interventions, including weight management and exercise. Given that there is currently no cure for osteoporosis, great emphasis should be placed on preventive strategies targeting such modifiable risk factors for osteoporosis. In the future, for the prevention and management of osteoporosis, more attention should be paid to the distribution of local adipose tissue rather than just focusing on total body weight or fat mass, with specific monitoring of VAT. Future research should continue to explore these relationships to better understand the mechanisms and implications of abdominal adipose tissue on the bone and to develop targeted interventions for high-risk populations.

The present study had several limitations. First, the study population consisted of living liver transplant donors who constituted a uniquely healthy cohort owing to the rigorous selection criteria. To ensure the effectiveness and safety of liver transplantation, donors must be free from major chronic diseases (e.g., no severe hypertension, diabetes, or cardiovascular disease), have normal liver function; have a BMI within a near-optimal range, and have no history of significant alcohol or substance abuse. Consequently, our cohort was inherently healthier than the general Chinese population at baseline. This selection bias likely influenced the findings of our study and limited their generalizability. Therefore, our results are most applicable for understanding these relationships in a healthy subset of the population and should not be overgeneralized. Future research validating these associations in general population-based cohorts that include individuals with a broader spectrum of health statuses and comorbidities is warranted. Secondly, this was a retrospective study; the association with bone fractures remains to be determined, and prospective studies are needed to evaluate their association with fractures, the most common clinical consequence of osteoporosis. Third, the cross-sectional design of the current study prevented us from directly evaluating the causal relationships between bone loss, BMAT increase, and changes in abdominal adipose tissue. Future longitudinal research should ideally assess whether BMAT affects bone quality and fracture risk. Fourth, lifestyle factors, hormonal variables, menopausal information, bone turnover biomarkers such as the N-terminal propeptide of type I procollagen and C-telopeptide of type I collagen were not available in the present study. Finally, given the relatively small number of participants, it is important to increase the sample size and include more diverse age groups to verify our findings. Further studies are required due to these limitations.

Conclusion

This study analyzed the relationships among BMAT, distribution of abdominal adipose tissue, and vBMD in a Chinese adult cohort. Our findings indicated that older age and higher BMAT, VAT, and TAT were correlated with reduced vBMD. A significant association was observed between BMAT and BMD. The impact of abdominal adipose tissue distribution on BMD varied; SAT exhibited a positive correlation with BMD, whereas VAT was negatively associated. These results contribute to growing evidence suggesting that the distribution of local adipose tissue, particularly visceral adipose accumulation, should be prioritized over total body weight in considerations of bone health.

Supplemental Information

Supplemental Information 1 The baseline characteristics and quantification values of bone mineral density of study subjects

Supplemental Information 2 Codebook for categorical variables in raw data

Supplemental Information 3 STROBE Checklist

The authors are very grateful to the participants who kindly contributed to this study.

Abbreviations

vBMD volumetric bone mineral density

QCT quantitative computed tomography

BMAT bone marrow adipose tissue

SAT subcutaneous adipose tissue

VAT visceral adipose tissue

TAT total adipose tissue

DXA dual energy X-ray absorption

IR insulin resistance

MRI magnetic resonance imaging

MRS magnetic resonance spectroscopy

PDFF proton density fat fraction

ROI regions of interest

BMI body mass index

AST aspartate transferase

ALT alanine aminotransferase

ALP alkaline phosphatase

GLU fasting blood glucose

PINP N-terminal propeptide of type I procollagen

CTX-I C-telopeptide of type I collagen

IDEAL-IQ Chemical shift-encoded iterative decomposition of water and fat with echo asymmetry and least squares estimation

SE standard error.

Additional Information and Declarations

Competing Interests

Author Contributions

Human Ethics

Data Availability

The authors declare there are no competing interests.

Wei Zhu performed the experiments, prepared figures and/or tables, authored or reviewed drafts of the article, and approved the final draft.

Wen-Min Guan conceived and designed the experiments, prepared figures and/or tables, and approved the final draft.

Bo-Xin Chen performed the experiments, authored or reviewed drafts of the article, and approved the final draft.

Yi-Da Lu performed the experiments, authored or reviewed drafts of the article, and approved the final draft.

Jia Li analyzed the data, prepared figures and/or tables, and approved the final draft.

Xiao-Qing Yuan analyzed the data, prepared figures and/or tables, and approved the final draft.

Wei-Hua Li analyzed the data, prepared figures and/or tables, and approved the final draft.

Feng-Xia Yu analyzed the data, prepared figures and/or tables, and approved the final draft.

Jing-Yi Liu conceived and designed the experiments, prepared figures and/or tables, and approved the final draft.

Hong-Xia Yin conceived and designed the experiments, authored or reviewed drafts of the article, and approved the final draft.

Lin Wei performed the experiments, authored or reviewed drafts of the article, and approved the final draft.

Zhen-Chang Wang conceived and designed the experiments, authored or reviewed drafts of the article, and approved the final draft.

Zhi-Jun Zhu conceived and designed the experiments, authored or reviewed drafts of the article, and approved the final draft.

Peng Zhang conceived and designed the experiments, prepared figures and/or tables, authored or reviewed drafts of the article, and approved the final draft.

The following information was supplied relating to ethical approvals (i.e., approving body and any reference numbers):

This retrospective study was reviewed and approved by the Institutional Review Board and the Ethics Committee of Beijing Friendship Hospital (2024-P2-211-01).

The following information was supplied regarding data availability:

The raw data is available in the Supplemental File.

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
