# Peer review of "Association between bone marrow adipose tissue, abdominal adipose tissue distribution, and volumetric bone mineral density in the Chinese adult population: a retrospective cohort study"

_PeerJ, doi:10.7717/peerj.20446_

## Round 0.1 · original submission · Major Revisions

· Academic Editor

Major Revisions

**Language Note:** The review process has identified that the English language must be improved. PeerJ can provide language editing services - please contact us at [email protected] for pricing (be sure to provide your manuscript number and title). Alternatively, you should make your own arrangements to improve the language quality and provide details in your response letter. – PeerJ Staff

·

Basic reporting

-

Experimental design

1. The study population consists of 306 liver transplant donors, who are likely healthier than the general Chinese population due to stringent donor eligibility criteria (e.g., absence of chronic diseases, normal liver function). This selection bias limits generalizability. Please discuss donor criteria and compare with general population studies.

2. Smoking, alcohol, and other lifestyle factors are collected, but variables are absent from the statistical analyses. Did smoking or alcohol intake correlate with vBMD, VAT, SAT, or BMAT? Why were these variables excluded from the regression model?

3. The multiple linear regression model includes TAT, VAT, and SAT as independent variables, despite TAT being the sum of VAT and SAT. This creates a risk of multicollinearity, where predictor variables are highly correlated, potentially leading to unstable estimates, overadjustment, or overfitting (Tuan V Nguyen. Common methodological issues and suggested solution in bone research. Osteoporosis and Sarcopenia. 2020; 6(4):161-167). The authors should remove TAT from the regression model.

4. The study reports differences in vBMD, VAT, SAT, and BMAT between males and females. Do the associations between VAT, SAT, BMAT, and vBMD differ by gender or age group? Could menopausal status in females influence these relationships?

Validity of the findings

5. The manuscript inaccurately states, ‘This approach is particularly advantageous as it allows for the assessment of BMD during routine abdominal CT scans without increasing radiation exposure. Although DXA is a reliable tool for determining BMD and abdominal obesity, it is unfeasible to distinguish different types of abdominal adipose tissue. These claims are misleading:

• QCT involves higher radiation exposure (1-3 mSv) compared to DXA (0.001-0.01 mSv), even for opportunistic screening (Brett AD, Brown JK. Quantitative computed tomography and opportunistic bone density screening by dual use of computed tomography scans. J Orthop Translat. 2015 Sep 15;3(4):178-184).

• Modern DXA systems can distinguish SAT and VAT through whole-body scans, contrary to the authors’ claim (Maskarinec G, et al. Subcutaneous and visceral fat assessment by DXA and MRI in older adults and children. Obesity (Silver Spring). 2022 Apr;30(4):920-930). \

• The cost-effectiveness of QCT, MRI versus DXA is not addressed, despite DXA being the preferred method for routine bone and adipose tissue assessment due to lower cost and radiation.

Additional comments

6. Please ensure consistent PDFF terminology across Fig.4, Table 2, and Table 3

Reviewer 2 ·

Basic reporting

This is a well-structured manuscript. The authors used appropriate references.

Experimental design

This is a cross-sectional study looking into the association between:
1- bone marrow adipose tissue (BMAT) and volumetric bone mineral density, and
2- abdominal adipose tissue distribution and volumetric bone mineral density.
The methodology is sound, and statistical analysis is properly conducted. The research question is well defined, and the authors clearly state the objective of the study. The authors also introduce the problem; however, I suggest a clearer explanation of why it is particularly important to report associations between BMAT/ adipose tissue and vBMD in the Chinese population, i.e., why the Chinese population may differ in terms of bone health outcomes and fat distribution compared to other more studied populations.

The sample size included in the study is large (n=306), but the age range is too wide. Participants’ age ranges from 19 to 66 years old, which raises concerns regarding bone health “maturity”, as well as fat distribution and BMAT levels. Bone health, fat distribution, and BMAT levels of a 19-year-old are significantly different from a 35-year-old, which in turn are also significantly different from a 66-year-old. Even though the authors controlled for age on their regression models, it would be interesting to see the associations between BMAT, adipose distribution, and vBMD according to different age ranges that reflect different “bone maturity”. Therefore, in addition to the statistical analysis and results already shown, I suggest that the authors group participants in different age ranges. For example, authors could distribute their participants according to the following groups: 1) pre peak bone mass (which can include participants with ages ranging from 19 – 30 years-old); 2) peak bone mass (which can include participants with ages ranging from 31 – 35 years-old); 3) pre-menopausal (which can include participants with ages ranging from 36 – 55 years-old); 4) post-menopausal (which can include participants with ages 56 and above). This new way to look at the data will not replace the well-done statistical analysis the authors already conducted, but it will allow us to see how adipose distribution and BMAT change according to different “bone maturation periods”. It would also allow us to see how BMAT and fat distribution correlate with vBMD at different “bone maturity” stages. It would be interesting to see this visually in a graph.

Validity of the findings

The authors acknowledge the limitations of using a cross-sectional design. However, special attention should be given to how strongly the statements are made in relation to the interpretation of the results. For example, in the abstract, the authors state, “SAT exhibits a positive effect on BMD, whereas VAT has a detrimental effect” – special attention should be paid to using the word “effect” as no intervention was conducted, and therefore, a cause-and-effect relationship cannot be reported. Authors should change the language to “positive associated” or “negative associated”. The same is seen in the discussion section (line 304) and the conclusion section (line 335). Please revise the use of the words “detrimental effect” and “positive effect”.

In the discussion (line 293) authors suggest that CSE-MRI can assist clinicians in diagnosing bone health. Caution should be taken as, currently, MRI is not a diagnostic tool. The current state of the art in relation to the interpretation of MRI data on bone marrow adipose tissue is still very limited. There are no cut-off values that determine healthy fat levels. In accordance, care should be taken when suggesting that MRI can be used as a tool to assess bone health and determine the risk of bone loss. I suggest that authors revise this sentence (lines 293 – 297).

Additional comments

Please revise the language of this sentence (not clear): “Finally, as presented in Figure 1, there were 306 donors of living 119 liver transplant donors aged over 18 years without known medical conditions affecting bone 120 metabolism were included.” (line 118).

Please consider adding in the title the “volumetric bone mineral density” instead of bone mineral density.

Reviewer 3 ·

Basic reporting

This is a well-designed cross-sectional study investigating the associations between bone marrow adipose tissue (BMAT), abdominal adipose tissue distribution (VAT, SAT, TAT), and lumbar volumetric bone mineral density (vBMD) in Chinese adult living liver transplant donors. The authors use advanced imaging methods (QCT, CSE-MRI) to quantify both bone density and adipose compartments, which is a strength. The sample size (n=306) is relatively robust, and the study addresses an understudied population.

The results are relevant, particularly the finding that BMAT and VAT are negatively associated with vBMD, while SAT shows a positive association. These observations could inform future strategies for osteoporosis prevention and risk stratification.

However, there are several areas where the manuscript can be improved: clearer articulation of novelty, methodological clarifications, more cautious interpretation of findings, and refinement of the language.

Experimental design

The manuscript presents original primary research that fits well within the aims and scope of the journal, addressing a relevant and meaningful research question on the associations of BMAT, VAT, SAT, and vBMD in a relatively young Chinese population. The study clearly identifies a knowledge gap by focusing on a younger cohort and using both QCT and CSE-MRI, thereby contributing novel insights beyond prior work largely centered on older or postmenopausal groups. The investigation appears technically sound and was conducted to appropriate ethical standards, with informed consent and institutional approval noted. Methods are described in sufficient detail to allow replication, including subject selection, imaging protocols, and statistical analyses, though some areas would benefit from clarification (e.g., rationale for cut-offs used, reproducibility or quality control of imaging measurements, and whether adjustments for potential confounders such as diet or physical activity were considered). Overall, the experimental design is rigorous, but could be strengthened with more information on sample size justification and measurement reliability.

Validity of the findings

The findings are generally valid and supported by robust statistical analyses, with results presented clearly and linked to the original research questions. The data appear appropriately controlled and internally consistent. The conclusions are mostly well stated and aligned with the results, but some interpretations—particularly regarding the protective role of SAT—should be expressed with greater caution given the cross-sectional design and potential residual confounding. While the study contributes novel insights into BMAT, VAT, SAT, and vBMD in a younger population, replication in larger and more diverse cohorts would be valuable to confirm generalizability and enhance the impact of the work.

Additional comments

In addition to the points noted above, the manuscript would benefit from clearer emphasis on the clinical or translational implications of the findings, as this would strengthen the significance for both scientific and medical audiences. The discussion could be improved by more explicitly addressing limitations, such as the relatively small sample size, single-center recruitment, and absence of lifestyle or hormonal factors that may influence bone density and adiposity distribution. It would also be helpful to clarify how the present results could inform future preventative or therapeutic strategies for osteoporosis in younger populations. Finally, ensuring consistent terminology (e.g., abbreviations for BMAT, VAT, SAT, vBMD) and careful proofreading throughout will improve readability and professionalism.

---

## Round 0.2 · accepted · Accept

· Academic Editor

Accept

Thank you for revising your manuscript to address the concerns of the reviewers. Reviewers 1 and 2 now recommend acceptance and I am satisfied that the comments of reviewer 3 have been addressed. The manuscript is now ready for publication.

·

Basic reporting

No comment

Experimental design

No comment

Validity of the findings

No comment

Additional comments

The authors have addressed all the issues raised. he manuscript was updated accordingly. I believe the manuscript is now suitable for publication.

Reviewer 2 ·

Basic reporting

The authors addressed well all comments.

Experimental design

The authors addressed well all comments.

Validity of the findings

The authors addressed well all comments.